# Bioactive Phyto-Compounds with Antimicrobial Effects and AI: Results of a Desk Research Study

**DOI:** 10.3390/microorganisms12061055

**Published:** 2024-05-24

**Authors:** Silviya Mihaylova, Antoaneta Tsvetkova, Emiliya Georgieva, Desislava Vankova

**Affiliations:** 1Medical College, Medical University of Varna, 9002 Varna, Bulgaria; antoaneta.tsvetkova@mu-varna.bg (A.T.); emiliya.georgieva@mu-varna.bg (E.G.); 2Department of Social Medicine and Health Care Organisation, Faculty of Public Health, Medical University of Varna, 9002 Varna, Bulgaria; desislava.vankova@mu-varna.bg

**Keywords:** AI, essential oils, PAMPs, AMR, network pharmacology, public health

## Abstract

Resistance of microorganisms to antibiotics represents a formidable global challenge, manifesting in intricate public health ramifications including escalated mortality rates and augmented healthcare costs. The current efforts to manage antimicrobial resistance (AMR) are limited mainly to the standard therapeutic approaches. The aim of this study is to present and analyze the role of artificial intelligence (AI) in the search for new phyto-compounds and novel interactions with antimicrobial effects. The ambition of the current research study is to support researchers by providing summarized information and ideas for future research in the battle with AMR. Inevitably, the AI role in healthcare is growing exponentially. The reviewed AI models reveal new data on essential oils (EOs) as potential therapeutic agents. In terms of antibacterial activity, EOs show activity against MDR bacteria, reduce resistance by sensitizing bacteria to the action of antibiotics, and improve therapeutic efficiency when combined with antibiotics. AI models can also serve for the detailed study of other therapeutic applications of EOs such as respiratory diseases, immune diseases, neurodegenerative diseases, and oncological diseases. The last 5 years have seen an increasing application of AI in the search for potential plant sources to control AMR. For the time being, the application of machine-learning (ML) models is greater in the studies of EOs. Future attention of research teams may also be directed toward a more efficient search for plant antimicrobial peptides (PAMPs). Of course, investments in this direction are a necessary preface, but the excitement of new possibilities should not override the role of human intelligence in directing research processes. In this report, tradition meets innovation to address the “silent pandemic” of AMR.

## 1. Introduction

Resistance of microorganisms to antibiotics represents a formidable global challenge, manifesting in intricate public health ramifications including escalated mortality rates and augmented healthcare costs. The high consumption of antibiotics within the agricultural sector and healthcare systems has diminished the efficacy of antibacterial interventions, thereby critically undermining our capacity to control infectious diseases.

Essential components in the fight against antimicrobial resistance (AMR) include the rapid detection of the etiological agents of the infection, the effective identification of resistant bacterial strains, the selection of appropriate therapy, and the prevention of potential complications. Current efforts to manage AMR are limited mainly to standard therapeutic approaches. Despite numerous programs to reduce consumption and control prescriptions, the resistance to clinically used antibiotics remains widespread, and the number of bacterial pathogens presenting multidrug resistance continues to rise [1]. Therefore, novel strategies to manage AMR are a social necessity. Intensifying the invention of new phyto-compounds with antimicrobial effects could be a sustainable strategy to manage AMR.

The discovery of drugs from medicinal plants continues to provide new and important phyto-products for the management of various pathologies including cancer, infectious diseases, including malaria, cardiovascular diseases, and neurological disorders [2]. The phyto-compounds with antimicrobial effects presented here in the context of AI are essential oils (EOs). EOs are complex lipophilic mixtures of volatile secondary metabolites of natural origin. Biologically active compounds in EOs are aromatic molecules with proven antibacterial, insecticidal, antiviral, and antifungal effects [3,4]. The studies of naturally occurring essential oils from medicinal plants have increased extremely over the last years [5]. AI could speed up these processes and could be an efficient tool for combating AMR in various ways [6]. The unique functions of AI to self-learn through machine learning (ML) and to forecast developing predictive models have been applied in several scientific fields such as drug design, predicting bioactivity, and physical properties of natural molecules, becoming an integral part of the discovery process of new drugs without the need for experimental research.

A dominant paradigm in the development of new drugs is the concept of designing maximally selective ligands to act on specific targets. However, many effective drugs work by modulating multiple proteins rather than individual targets. Further, in 2007, the term “network pharmacology” was introduced [7]. Network pharmacology is an approach to drug design that encompasses systems biology, network analysis, connectivity, redundancy, and pleiotropy. Network pharmacology offers a way of thinking about drug discovery that simultaneously embraces efforts to improve clinical efficacy and understand side effects and toxicity—two of the most important reasons for failure [8]. Network pharmacology is intricately connected with AI and ML techniques, leveraging these advanced computational tools to analyze, predict, and simulate the complex interactions within biological systems, including EOs.

The aim of this study is to present and analyze the role of AI in the search for new phyto-compounds and novel interactions with antimicrobial effects. The ambition of the current research study is to support researchers by providing summarized information and ideas for future research in the battle with AMR.

## 2. Materials and Methods

A desk review was performed using an electronic search encompassing PubMed database, over a 10-year period (April 2015–March 2024). The search strategy involved the primary keyword “essential oils” coupled with “artificial intelligence” and “network pharmacology” (Figure 1 and Figure 2), and “antimicrobial resistance” to identify relevant literature.

## 3. Results

The results are structured in two basic parts according to the two electronic searches that combine EO with AI or with network pharmacology in the context of AMR.

There are 62 publications found in PubMed under the keywords “artificial intelligence” and “essential oils”, and 54 of them were published after 2015. Only six of them [9,10,11,12,13,14,15] (Daynac, 2015; El-Attar, 2020; Artini, Papa, Ragno, 2019 and Patsilinakos, 2020) considered the direct application of AI to predict the antimicrobial activity of EOs in different aspects (Figure 1). When “antimicrobial resistance” was added to these two keywords, four publications were found for Eos that overlap with those already listed in Figure 1.

### 3.1. AI and Eos

Given the multicomponent composition of Eos, it is difficult to identify an exact or specific antibacterial mechanism of action. Due to their lipophilic nature, most often Eos penetrate the cell wall, disrupt the integrity of the cell membrane, and induce cell death. The use of artificial intelligence to determine the active antibacterial components of Eos is a fast and reliable approach in experimental studies. Daynac et al. apply the artificial neural network (ANN) model to select active compounds responsible for the antimicrobial action of essential oils (Eos) against *S. aureus*, *E. coli*, *C. albicans*, and *Cl. Perfringens* [9]. The authors used the real zones of inhibition obtained in vitro by the disk diffusion assays to determine the accuracy of the predictive models. Based on the difference between the in silico data and the real inhibitory diameters (ΔID), the prediction accuracy was determined as follows: ΔID ≤ 5 mm—very accurate prediction, ΔID ≤ 10 mm—accurate prediction, and ΔID > 10 mm—wrong prediction. The model achieves an accuracy of > 70% of the ANN prediction within a maximum error range of 10 mm. The algorithm developed can also take into account synergistic and antagonistic interactions between compounds.

AI has the potential to discover the relationship between the biological activity of plant compounds and their chemical composition. El-Attar et al. [10] compared two ML models, the multiclass neural network (MNN) and convolutional neural network (CNN), to predict the biological activity of a set of endemic Egyptian plants rich in EOs. The results obtained show a better prediction accuracy using the deep learning algorithm (CNN) compared to the machine-learning algorithm (MNN): 98.13% and 81.88%, respectively. According to the authors, CNN is a reliable AI algorithm for predicting the bioactivity of Egyptian plants.

Four of the publications (Artini, Papa, Ragno, and Patsilinakos) included in this desk review are from a team of scientists from Sapienza University. Originally, Artini et al. [11] and Patsilinakos et al. [12] used ML classification techniques to develop a quantitative activity–composition relationship (QCAR) to discover the chemical components mainly responsible for antibiofilm activity. They analyzed the anti-biofilm activity of 89 samples of different essential oils against *P. aeruginosa*, *S. aureus* ATCC 6538P, *S. aureus* ATCC 25923, *S. epidermidis* RP62A, and *S. epidermidis* O-47. Using the antibacterial activity results from in vitro experiments, Artini et al. applied an in-house machine-learning protocol based on Python and defined a classification model capable of dividing essential oils into two groups (inhibiting and potentiating) biofilm formation (*P. aeruginosa*) using a concentration of 48.8 µg/mL.

Following the work of Artini et al., Patsilinakos et al. used the same set of EOs and investigated their potential antibacterial and antibiofilm effects against *S. aureus* ATCC 6538P, *S. aureus* ATCC 25923, *S. epidermidis* RP62A, and *S. epidermidis* O-47. The application of principal component analysis (PCA) coupled with logistic regression led to the formulation of robust models against four pathogens, which were validated by four coefficients: accuracy, MCC (Matthews correlation coefficient), precision–recall, and ROC-AUC (area under the receiver operating characteristic). ML models demonstrate biofilm activation in both *S. aureus* strains, and biofilm inhibition in both *S. epidermidis* strains [13].

Papa et al. [14] reported the biofilm production modulation exerted by 61 EOs, and they also investigated their antibacterial activity on *S. aureus* strains, including reference and cystic fibrosis patients’ isolated strains. The chemical composition, investigated by GC/MS analysis, of the tested EOs allowed a correlation between the biofilm modulation potency and putative active components by means of ML algorithms applied to develop classification models. Some EOs inhibited biofilm growth at a concentration of 1.00%, although lower concentrations revealed a different biological profile.

Ragno et al. [15] investigated the activity of 61 essential oils (EOs) against 40 bacterial strains of *S. aureus* and *P. aeruginosa* isolated from cystic fibrosis (CF) patients. The authors chose these two pathogens because they are associated with the worst clinical prognosis. To reduce the time for in vitro studies, the authors initially used unsupervised machine-learning algorithms and techniques, as implemented in Python language 13. Using phenotypic and genotypic characteristics as categorical descriptors, ML algorithms selected nine clinical bacterial strains—six for *P. aeruginosa* and three for *S. aureus*. The antibacterial activity of all 61 EOs was tested on the 9 clinical isolates and reference strains selected already, determining the minimal inhibitory concentration (MIC). Of all the tested EOs, the highest antibacterial activity was shown in three EOs (cade, birch, and Ceylon cinnamon), which were then studied for activity against all clinical isolates. The results confirmed that they exerted a strong and effective bactericidal potency on all tested clinical strains. After these results, the authors investigated the likely chemical components of the three EOs mainly responsible for antibacterial activity, using GC/MS analysis.

### 3.2. Network Pharmacology and EOs

The pharmacological efficacy of multicomponent systems increases as a result of intramolecular interactions, resulting in a clinically significant synergistic effect. Network pharmacology is a promising method for elucidating the relationship while analyzing the results from a holistic perspective [16].

By keywords “essential oils” and “network pharmacology”, the number of publications is 139, of which 92 were published after 2020, and 43 are open access (Figure 2). One-third of these publications addressed other than antimicrobial indications for EOs (depression, atopic dermatitis, Alzheimer’s disease, allergic rhinitis, etc.).

Buriani et al. [17] investigated the cytotoxic activity of EOs from various species of the *Pistacia* genus on human tumor cell lines by combining the results of GC/MS analysis and principal component multivariate analysis (PCA). The biological activity of the different samples was determined against the EC molecular fingerprint. PCA analysis shows the contribution of each sample compound to the cytotoxic effect of the phyto-complex. It is important to note that the analysis does not take into account the contribution of individual molecules but the end result of their presence in the biological environment, providing an inductive and at the same time holistic reading of the experimental evidence. Merging the chemical composition data and the biological results by a multivariate approach allows for the evaluation of the bioactivity of complex mixtures.

Using an integrated network pharmacology approach, You et al. [18] described various mechanisms of action and pharmacological effects of essential oil from the leaves of *C. grandis* ‘*Tomentosa*’. The qualitative and quantitative composition of the essential oil was determined by gas chromatography coupled with tandem mass spectrometry (GC-MS/MS). The targets of 61 compounds were predicted and selected from different databases such as the Traditional Chinese Medicine Systems Pharmacology Database and Analysis Platform (TCMSP, http://tcmspw.com/tcmsp.php, accessed on 1 March 2024), SwissTargetPrediction (http://www.swisstargetprediction.ch/, accessed on 4 March 2024), STITCH (http://stitch.embl.de/, accessed on 28 February 2024), and Similarity Ensemble Approach (SEA, http://sea.bkslab.org/, accessed on 1 March 2024). The Search Tool for the Retrieval of Interacting Genes (STRING) database was used to construct the protein–protein interaction (PPI) networks. After filtering by condition, four direct targets were found to be highly correlated with the composition of essential oils. The genes PTGS2, CHRM1, GGPS1, and MAPK14 were associated with (−)-γ-cadinene, bicyclogermacrene, β-elemene, and β-citronellol, respectively.

*Streptococcus mutans* is a naturally occurring microorganism in the human oral microbiota and its overprevalence is associated with the development of caries. Yuan et al. [19] investigated the antibacterial activity of two different oregano essential oils (OEOs) against *S. mutans* using in vitro methods (disk diffusion method, minimum inhibitory concentration, and minimum bactericide concentration). The results show that OEOs in a concentration below the MIC suppresses the growth of *S. mutans* and inhibits its cariogenic activity (evaluated by a glycolytic pH drop assay). The authors determined the chemical composition of OEOs by GC-MS, and the interaction of key compounds (carvacrol and its biosynthetic precursors γ-terpinene and p-cymene) with the target virulence proteins of *S. mutans* was predicted by molecular docking. Using efficient network pharmacology analysis, the authors identified potential key compounds with antibacterial activity and determined their synergistic effects in the multicomponent composition of OEOs. Yuan et al. also investigated the cytotoxic potential (MTT test) of EOs on immortalized human keratinocyte (HaCaT) cells. The results showed that OEOs did not induce toxic effects at concentrations of 0.1 μL/mL in HaCaT cells.

Most studies have investigated the antimicrobial effects of whole-composition compounds of essential oils rather than the activity of the pure compounds. It is believed that the antimicrobial activity of EOs is due to the chemical compounds that are the most abundant, without clear evidence for this. Carev et al. [20] through in silico correlation analysis and experimental approaches investigated the relationship between the chemical constituents found in *C. triumfettii* essential oil and their antimicrobial activity in order to identify the potential active chemical components. After a literature review, the authors selected six chemical compounds (germacrene D, aromadendrene, spathulenol, longifolene, linoleic acid, and hexadecenoic acid), which have the highest content in the essential oil of *C. triumfettii*. The antibacterial activity of the target chemical compounds, both as pure compounds and as a mixture of selected components, on *E. coli* and *S. epidermis* strains was determined by the agar diffusion method. The experimental results do not indicate that any of the pure components in the essential oil are related to its antibacterial activity. The network pharmacology analysis used by the authors also revealed a complex network of interactions between the individual components, and it was not possible to identify only one specific chemical compound responsible for the antimicrobial activity. The results suggest that the antibacterial effect of essential oils may be partially related to a synergistic effect between the individual components and also to different mechanisms of action such as disruption of the lipid fractions of bacterial plasma membranes and fatty acid metabolism.

Zhao et al. [21] investigated the antibacterial effect and detailed the mechanism of action of menthone (the main component of *Mentha piperita* essential oil) against methicillin-resistant *Staphylococcus aureus* (MRSA). The authors took an integrated approach involving high-coverage microbiology and lipidomics. Through network analysis and molecular docking, the lipid targets associated with the inhibitory effect of menthone against MRSA were determined. The results of in vitro tests show that menthone exhibits a strong bactericidal effect against the MRSA strain. The antibacterial mechanism of action established by the authors includes, on the one hand, the depolarization of the bacterial membrane and its destruction, and on the other hand, the disruption of the lipid homeostasis of MRSA cells.

Using graph embedding, Yabuuchi et al. [22] developed a machine-learning method that predicts synergistic or antagonistic interactions between combinations of two different essential oils. The performance of the in silico method was evaluated by cross-validation, a statistical method for evaluating learning algorithms. Using the classifier constructed, the authors studied the probability of synergistic/antagonistic interaction between all possible pairs of the commercially available 84 EOs. Initially, the classifier predicted 2088 EO pairs as synergistic, from which the authors randomly selected 16 EO pairs for following gas chromatography/mass spectrometry (GC/MS) analysis and in vitro antibacterial assay. When comparing the results obtained by antibacterial assay, the precision of ML models was apparently low (25% (4/16)), but this result indicates that the proposed approach is applicable to a wide variety of EOs. The model can be further developed and refined for higher accuracy.

## 4. Discussion

The greatest application and impact of AI is expected to be in education and in healthcare including research for health [23]. The presented desk review and analysis are supported by the research experiences of the authors in the fields of EOs and AMR [24], and the One Health strategies to manage AMR [25]. The professional interest was provoked by the experiences as educators in the new university programs, AI in Healthcare and AI in Biomedicine.

The development and introduction into clinical practice of new antibacterial therapies are not enough to manage the growing threat of AMR [26]. In the period from 1935 to 2003, fourteen new classes of antibiotics were introduced into clinical practice, while only ten new antibiotics were approved since 1998, of which two (linezolid and daptomycin) had a new mechanism of action [27]. According to a World Health Organization (WHO) report containing antibiotic production data for 2021, eleven new antibiotics have been approved since 2017. Only two of them represent a new class and have a new target of action (vaborbactam + meropenem and lefamulin) [28]. As of 2024, a new molecule—zosurabalpin—is in phase I clinical trials. It belongs to a new class of antibiotics called tethered macrocyclic peptide antibiotics, which show activity against CRAB (carbapenem-resistant *Acinetobacter baumanni*) [29].

Evidently, CAM therapies like phytotherapy can provide solutions contributing to managing the overuse and misuse of antibiotics. Finding novel effective antibacterial compounds against multiple drug resistance (MDR) bacteria is a global health priority. The WHO Traditional Medicine Strategy 2014–2023 highlights involving traditional, complementary, and integrative healthcare (TCIH) in national public health strategies and national action plans (NAPs) on AMR as options in the prevention and treatment of infectious diseases, and reducing antibiotic use [30]. In a review article, Álvarez–Martínez et al. [31] describe scientific literature published between 2016 and 2020 related to the antibacterial activity of compounds of plant origin, with special emphasis on their activity against MDR bacteria and their mechanisms of action. The authors indicate that for the period under review, the most studied antibacterial agents were plant extracts. EOs, as multicomponent mixtures, are in fifth place in the number of publications. However, with respect to pure compounds, for the period indicated by the authors, the greatest number of studies have focused on the antibacterial activity of terpenes, which are one of the main components of EOs, followed by flavonoids and alkaloids. The antimicrobial activity of essential oils is often associated with their terpene compounds. Some phytochemicals have the ability to inactivate or weaken antibiotic resistance mechanisms by sensitizing bacteria to the action of antibiotics. This ability leads to a synergistic action between certain phytochemicals and antibiotics, the efficacy of which would be very low in the absence of phytochemicals. Some of the phytochemicals are not active when used alone and show appropriate activity only when administered together with an antibiotic [31].

The One Health concept, while not a novel paradigm, has seen a significant escalation in its importance in recent years, underscored by the advent of recent epidemics and pandemics, notably COVID-19. This has elucidated the complex interdependence between human, animal, plant, and environmental health, advocating against their isolated consideration. Concurrently, there has been a discernible surge in antimicrobial resistance (AMR) and in the prevalence of zoonotic diseases (including COVID-19, avian influenza, and mpox) within the European region. Empirical evidence substantiates that identical pathogens proliferate amongst humans and animals, culminating in analogous pathologies, primarily exacerbated by the homogeneous application of antibiotics across species. This homogeneity in treatment methodologies fosters an environment conducive to the amplification of AMR. In a preemptive measure, the European Union (EU) instituted a comprehensive prohibition on the utilization of antibiotics as feed additives for animals as early as 2006. Subsequently, in June 2017, the European Commission ratified the European One Health Action Plan against AMR [32]. An analytical review conducted in 2023 delineated the impediments to the efficacious enactment of AMR countermeasures within EU Member States, Norway, and Iceland [33].

The academic and research community increasingly acknowledges the One Health concept as an instrumental public health ideology, pivotal for its holistic influence on the myriad facets of life and its capacity to unify disparate sectors globally. One Health has the potential to preserve and even save planetary health. The contemporary academic and research objective is to sustain the efficacy of antimicrobial therapies for both humans and animals. Specifically, the One Health research and education strategies are evolving across several vectors:Endorsement of AMR and the One Health initiative as fundamental responsibilities of academic educators by providing appropriate training and raising community awareness about the judicious use of antibiotics.Augmentation of interdisciplinary collaboration, fostering innovative methodologies and instruments for the prophylaxis and management of infectious maladies.Expansion of expertise in efficacious infection control strategies, encompassing novel diagnostic techniques.Exploration and development of innovative or alternative therapeutic agents and vaccinations.Integration of holistic medical approaches to diminish antibiotic prescription dependency.Provision of expert recommendations for legislative amendments to restrict antibiotic sales.Establishment of a cross-sectoral One Health framework to facilitate coordination among veterinary, food, and health regulatory bodies.Implementation of comprehensive surveillance mechanisms for AMR and antimicrobial utilization at the community and hospital levels.

The reviewed AI models reveal new data on EOs as potential therapeutic agents. Clearly, AI supports scientists in researching the role of EOs in combined antimicrobial therapies. The last 5 years have seen an increasing application of AI in the search for potential plant sources to control AMR. For the time being, the application of ML models is greater in the studies of EOs. Future attention of research teams may also be directed toward a more efficient search for plant antimicrobial peptides (PAMPs).

The initial idea for keywords also included PAMPs and AI. Analogously to the searches for EOs in PubMed, we also looked at publications on the keywords “artificial intelligence” and “plant antimicrobial peptides”. However, the results showed only sixteen publications, seven of them full-text publications. Of these seven publications, there is none that addresses the discovery of PAMPs using AI. Currently, with the help of AI models and algorithms, potential plant-derived peptides are considered mainly as antiviral agents [34,35,36], and genes for resistance or plant reproduction are studied [37,38]. Lu et al. [39] used network pharmacological analysis and molecular docking to predict key compounds, key targets, and binding energies based on the detected compounds in *Jasminum grandiflorum* EO, for treating AD by inhibiting microglia.

In terms of synergism with traditional antibiotics, the combination most often studied for synergism is phytochemicals with beta-lactam antibiotics. The most common mechanisms of synergistic action are the inhibition of efflux pumps, followed by the inhibition of beta-lactamases and increased membrane permeability. Various supervised ML models are also used to predict synergistic effects between AMPs and approved antimicrobials. Olcay et al. [40] established an advanced ML-based computational technique (oLGBMC) that identifies synergistic antimicrobial effects between different AMPs and chemical drugs based on the checkerboard assay principle and FIC index calculation. The results showed that the hyperparameter-optimized light gradient enhanced machine classifier (oLGBMC) gave the best test accuracy of 76.92% for synergistic prediction and enabled its future use in searching for combinatorial approaches to address microbial infections.

## 5. Conclusions

Technological breakthroughs and digital approaches inevitably lead to an exponential increase in the role of AI in healthcare. This review began with a brief presentation of therapeutic options in the treatment of bacterial infections in the context of AMR. In addition to the difficulties experienced by healthcare professionals in treating bacterial infections caused by resistant microorganisms, AMR is also associated with high economic and societal costs. In recent years, there has been a growing interest in studying natural products as a source of potential compounds that can be used in the development of new antibacterial drugs. The aim was achieved to consider the possible applications of AI for a more effective search for new bioactive substances of plant origin with antimicrobial activity, with an emphasis on EOs. Certainly, investment in this direction is a necessary prelude, but the excitement of new possibilities should not override the role of human intelligence in guiding research processes. In this report, tradition meets innovation to tackle the “silent pandemic” of AMR.

## Figures and Tables

**Figure 1 microorganisms-12-01055-f001:**
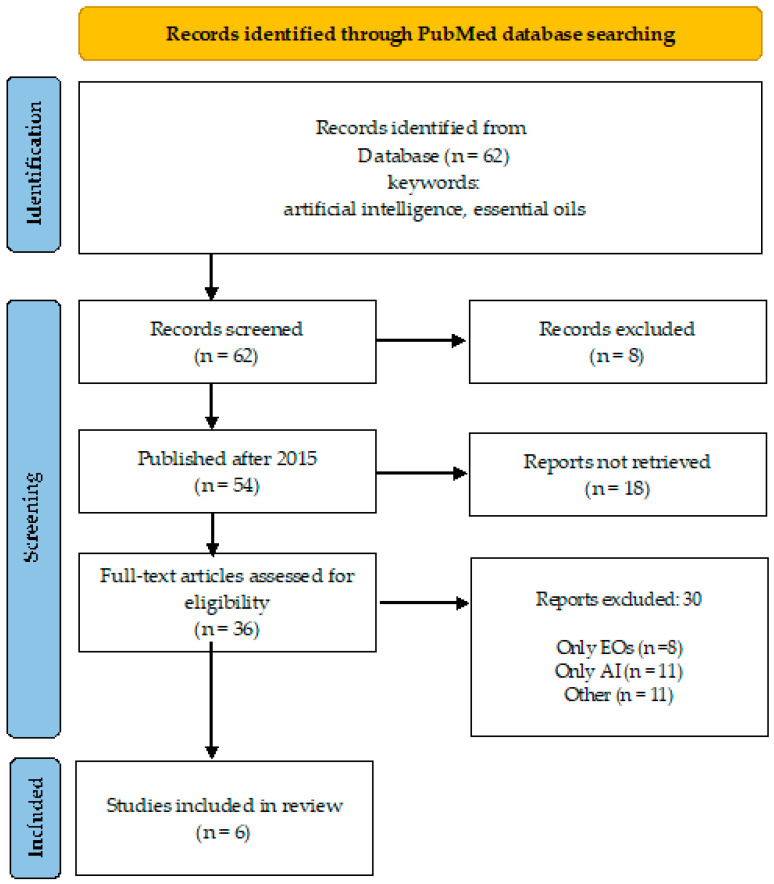
Overview of the search process (keywords: artificial intelligence, essential oils).

**Figure 2 microorganisms-12-01055-f002:**
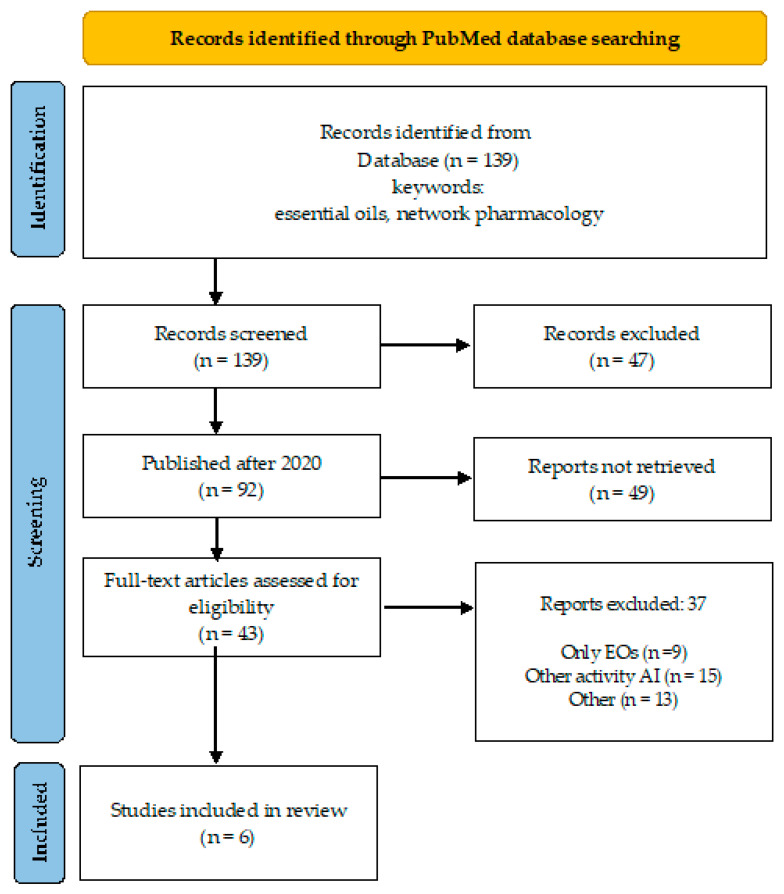
Overview of the search process (essential oils, network pharmacology).

## Data Availability

All data generated or analyzed during this study are included in this published article.

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
