# Peer review of "Bioactive Phyto-Compounds with Antimicrobial Effects and AI: Results of a Desk Research Study"

_microorganisms, 2024, doi:10.3390/microorganisms12061055_

Round 1

Reviewer 1 Report

Comments and Suggestions for Authors

The manuscript "Bioactive Phyto-Compounds With Antimicrobial Effects And AI: Results Of A Desk Research Study" is interesting, but has a lot of abbreviations without explanation!

In the abstract, state the meaning of the abbreviations: AMR, AI, instead of keywords!

Explain why the period January 2015 - March 2024 was taken for analysis!?

What does the abbreviations MCC and ROC-AUC mean? (Line 125, page 4)

Moreover, explain TCMSP, STITCH, SEA and STRING...

Reviewer 2 Report

Comments and Suggestions for Authors

The review presents an analysis of publications devoted to the use of artificial intelligence in assessing the antimicrobial potential of biologically active compounds of natural origin.

MAIN CRITICISM

The authors make the main idea of the work AI analysis of antibacterial components from natural sources, but in the results section other effects and diseases are mentioned, which distracts the attention of readers. For example,

The component-target-disease net- 179

work diagram revealed that the essential oil compositions could treat tumors, immune 180

diseases, neurodegenerative diseases and respiratory diseases, which were highly related 181

to genes CHRM1 (cognitive impairment; chronic obstructive pulmonary disease), PTGS2 182

(pain; arthritis) , and three genes three genes associated with solid tumor/cancer (CASP3, 183

MAP2K1 and CDC25B

In the discussion, I would like, among other things, to see conclusions regarding which neural network is most promising in relation to the analysis of the most biologically active natural components.

Figure 1. Overview of the searching process (keywords: artificial intelligence, essential oils), Figure 2. Overview of the searching process (essential oils, network pharmacology). – in my opinion, not very informative

Line 97 achieves accuracy >70% of the ANNs prediction within a maximum error range of 10 mm – what do you mean? This value should be explained

 MINOR REMARKS

Line 17 The reviewed AI models reveal new data on EOs as potential therapeutic agents. The last 5 - decipher

Line264 egy 2014–2023 hightlight involving TCIH in national public health strategies and National - decipher

Line 239rial assay. При сравнение на резултатите, obtained by antibacterial assay, the precision - ?

Species names should be noted in italics.

Comments on the Quality of English Language

Moderate editing

Round 2

Reviewer 2 Report

Comments and Suggestions for Authors

I am quite satisfied with the authors’ answers and the work done to improve the manuscript. However, based on the data presented, a conclusion should still be drawn about the most promising network that can evaluate antibacterial effectiveness. I leave this comment to the discretion of the Editor-in-Сhief

Comments on the Quality of English Language

Moderate editing